# Application of Cyclodextrin for Cancer Immunotherapy

**DOI:** 10.3390/molecules28145610

**Published:** 2023-07-24

**Authors:** Xiaojie Wei, Cui-Yun Yu, Hua Wei

**Affiliations:** Hunan Province Cooperative Innovation Center for Molecular Target New Drug Study, School of Pharmaceutical Science, University of South China, Hengyang 421001, China

**Keywords:** cyclodextrin, cancer, immunotherapy

## Abstract

Tumor immunotherapy, compared with other treatment strategies, has the notable advantage of a long-term therapeutic effect for preventing metastasis and the recurrence of tumors, thus holding great potential for the future of advanced tumor therapy. However, due to the poor water solubility of immune modulators and immune escape properties of tumor cells, the treatment efficiency of immunotherapy is usually significantly reduced. Cyclodextrin (CD) has been repeatedly highlighted to be probably one of the most investigated building units for cancer therapy due to its elegant integration of an internal hydrophobic hollow cavity and an external hydrophilic outer surface. The application of CD for immunotherapy provides new opportunities for overcoming the aforementioned obstacles. However, there are few published reviews, to our knowledge, summarizing the use of CD for cancer immunotherapy. For this purpose, this paper provides a comprehensive summary on the application of CD for immunotherapy with an emphasis on the role, function, and reported strategies of CD in mediating immunotherapy. This review summarizes the research progress made in using CD for tumor immunotherapy, which will facilitate the generation of various CD-based immunotherapeutic delivery systems with superior anticancer efficacy.

## 1. Introduction

Cancer remains one of the major threats to human health worldwide, with a low cure rate, high recurrence, and metastasis. There are 14 million new cases and more than 80 million deaths from cancer every year [1]. Compared with other treatment methods, chemotherapy and radiotherapy are still the mainstream clinical treatment strategies for cancer. However, the effects of chemotherapy and radiotherapy in preventing tumor metastasis and recurrence remain unsatisfactory [2,3]. Tumor immunotherapy aims to achieve the antitumor effect by harnessing the body’s innate immune system. The innate immune system consists mainly of epithelial barriers, granulocytes and macrophages, dendritic cells, and natural killer cells [4,5]. Immunotherapy can simultaneously mitigate tumor metastasis and recurrence [6]. The current cancer immunotherapy primarily encompasses monoclonal antibodies, cancer vaccines, cytokines, and immune checkpoint blockade therapy. The efficacy of cancer immunotherapy relies on the active participation of various immune cells. Tumor-associated antigens are recognized initially by antigen-presenting cells (APCs), with dendritic cells (DCs) and macrophages being particularly vital. Within the human leukocyte antigen (HLA) class I and II complexes, DCs present tumor-associated antigens to CD8+ T cells (CTL) and CD4+ helper T cells (Th), respectively [7,8,9,10]. Subsequently, the Th1 and Th2 subtypes, stimulated by tumor-associated antigens, can further activate CTL by releasing cytokines like interferons (IFNs) and interleukins (ILs) [11]. These activated immune cells, together with innate immune cells like natural killer cells (NKs) and natural killer T cells (NKTs), accumulate at the tumor site to exert potent antitumor effects [12].

However, the efficacy of immunotherapy can be attenuated by various influential factors. There are probably three reasons: (1) the poor water solubility of immunotherapy drugs, which affects the efficacy; (2) immunosuppression; and (3) immune escape [13]. The innate immune system has an immunosurveillance role for recognizing and further destroying tumor cells. However, it has been shown that tumor cells will utilize immune escape in order to “survive” and eliminate highly immunogenic clones by creating an immunosuppressive tumor microenvironment, which leads to a significantly compromised killing effect of immune systems on tumor cells and even promotes growth and metastasis of tumor cells. That is probably why the immune system is defined as a double-edged sword to some extent. Therefore, there is considerable scope to improve the tumor-suppressive microenvironment and reverse the immune escape properties of tumor cells for enhanced immunotherapeutic efficiency [14,15,16]. Tumor-associated macrophages (TAMs) are considered to be the main participants in the tumor microenvironment. TAMs account for over half of tumor immune cells and are involved in almost all stages of tumor progression. Additionally, tumor macrophages have been proven to be associated with immunotherapy resistance [17].

To address these limitations of immunotherapy, our proposal entails mitigating these drawbacks via the utilization of polymer materials. Among these materials, cyclodextrin has garnered significant attention in the realm of polymer research owing to its unique structural characteristics, which have found broad applications in enhancing various therapeutic species.

To this end, we conducted a comprehensive summary on the use of CD for cancer immunotherapy, which distinguishes this review from most published papers with a similar research subject. Specifically, this review focused on the delivery of a single immune-modulator via CD-based carriers for cancer immunotherapy, as well as further integration of immunotherapy with other therapeutic modalities. Overall, this review aims to inspire the future development of advanced CD-based nanocarriers for efficient tumor immunotherapy.

## 2. Current Status of Cyclodextrins

### Properties of Cyclodextrins

CDs are cyclic structures consisting of multiple glucose units linked together [18,19,20,21,22]. Depending on the number of linked unit structures, generally six, seven or eight units, they are classified as α-, β-, and γ-cyclodextrins, respectively (Figure 1 and Table 1). The slightly different properties of three typical CD members are discussed briefly as follows. α-CD has good water solubility and can only host guest substances with a relatively small dimension that fits the cavity of α-CD, β-CD is insoluble in water and can include guest substances with medium molecular weights in the cavity, and γ-CD has the best water solubility and can contain a wider range of guest substances [23]. Among these three CDs, β-CD has been probably the most frequently used pharmaceutical excipient. β-CD does not easily decompose in the human stomach but can undergo biodegradation in the gut by digestive enzymes and intestinal microbes to participate in human metabolism as a common carbohydrate without any accumulation effects. Furthermore, the chief reason why β-CD has become probably the most popular candidate among the three CDs is that β-CD’s geometry, consisting of seven glucose units, is the most suitable architecture for forming inclusion complexes with currently developed drugs. α-CD’s openings on the ring surface are insufficient to accommodate many drugs, and γ-CD, with a relatively large cavity, is thought to be unfavorable for constructing stable inclusion complexes. To improve the hydrophilicity of β-CD for hosting hydrophobic anticancer drugs, various β-CD-based derivatives have been developed, including hydroxypropyl-β-cyclodextrin (HP-β-CD) and sulfonic acid derivatives of β-CD [24].

CDs generally integrate a hydrophobic cavity and a hydrophilic outer surface, which enables the inclusion of various hydrophobic bioactive agents with an appropriate size in the cavity for enhanced solubility and bioavailability via supramolecular host–guest interactions [25,26].

The formation of a CD/drug complex is an entropy-driven process, with hydrophobic interactions as the primary driving force [27]. The following forces are also involved: van der Waals forces, hydrogen bonding, electrostatic interactions, hydrophobic interactions, etc. In addition, CDs have advantages such as easy functionalization, biocompatibility, biodegradability and low immunogenicity. Moreover, the hydroxyl groups on the primary and secondary faces of CDs can be further modified by various chemistry strategies for tailor-made functionalities (Figure 2) [28].

In addition to natural polysaccharides, synthetic peptides represent another type of important bio-inspired material for constructing controlled delivery systems (DDS) due to a precisely modulated sequence of amino acids for tailor-made properties that can mimic the various functions of natural materials for improving anticancer delivery efficiency. An elegant integration of synthetic peptides and natural polysaccharides provides a robust strategy for achieving greater anticancer therapeutic effects [29,30,31].

## 3. Application of Cyclodextrins in Cancer Immunotherapy

### 3.1. Cyclodextrins as Immunomodulators 

Zhao Q et al. conducted a study on the impact of CD analogues as immunomodulators on oligonucleotide-induced immune stimulation both in vivo and in vitro. They utilized the [3H] thymidine incorporation test to evaluate the effect of incubating spleen cells with 27 different polyphenyl thiophosphate oligonucleotides, known to induce immune stimulation, on cell proliferation. The results showed that the combination of CD analogues and thiophosphate oligonucleotides significantly reduced the cell proliferation induced by the oligonucleotides. Furthermore, administration of these 27 polyphosphate oligonucleotides in mice led to splenomegaly and increased production of IgM after 48 h. However, when oligonucleotides and CD analogues were administered together, they effectively inhibited splenomegaly and the IgM response. The degree of inhibition depended on the concentration of CD analogues, and similar observations were made with other immunostimulating thiophosphate oligonucleotide sequences. Notably, cyclodextrin analogues alone did not have any effect on splenomegaly or immune stimulation [32]. 

### 3.2. Cyclodextrin Host–Guest Recognition, Loading Small Molecule Immunomodulators 

Based on the unique structural characteristics of CD, host—guest complexation is the main strategy for the production of complexes based on CD and pharmaceutical preparations. In the context of prostate cancer treatment, Sun’s silicic acid-bound CD/CSF-1R siRNA supermolecule nanoparticles can target the siglec-1 (CD169) overexpressed by M2 macrophages [33] to obtain the up-regulated CSF-1/CSF-1R signaling pathway [34], which is related to poor prognosis of cancer patients [35] (Figure 3).

The CSF-1/CSF1R signaling pathway is the main target of tumor-associated macrophages (TAMs). The high expression of this pathway in M2 macrophages is associated with poor prognosis in cancer patients. Silicic acid seems to be a targeted ligand for overexpression of siglec-1 (CD169) on M2 macrophages. Therefore, they conceived the silica-bound CD/CSF-1R siRNA supermolecule nanoparticles.

For the application of CD in tumor macrophages, Rodell et al. proposed a TAM re-education strategy. This strategy has unlimited potential in combination with immune checkpoints, in which CDs play an important role that cannot be ignored. The systematic TAM targeting properties, due to the glycopolymeric nature, good biosafety and degradability of cyclodextrins, have made CDs an excellent candidate. The team loaded the TLR 7/8 agonist with the best re-education TAM effect onto cyclodextrin by host–guest recognition to form nanoparticles and deliver them to the body to exert immunotherapy [36].

In the field of tumor vaccines, Zhang et al. reported a supramolecular assembly programmable nanodrug (PIAN) as an in situ cancer vaccine. PIAN is fabricated via β-CD/Ad host–guest interactions among PPCD, mPEG TK Ad (PEG thione adamantane) and CpG/polyamide amine thione adamantane (CpG/PAMAM-TK Ad). After being injected intravenously into the body, the vaccine first circulates in the blood and then flows through tumor tissue. Then, accumulation begins in the tumor tissue through the EPR effect. Then, due to the high ROS of the tumor microenvironment, CpG/PAMAM and PEG are separated. The ingestion of PPCD in vaccines by tumor tissue leads to tumor cell death and antigen release. The released antigen is captured by CpG/PAMAM, forming a complex, and then reached the tumor-draining lymph nodes (TdLNS). Then, dendritic cells (DC) internalize CpG/PAMAM/antigens, leading to activation within the DCs. Activated DCs present antigens to T cells, and ultimately tumor-specific effector T cells return to the tumor site to exert anti-tumor effects [37] (Figure 4).

Yang et al. prepared a combined hydrogel/nano adjuvant-mediated cellular vaccine for cancer immunotherapy. This vaccine combines tumor cells and DC vaccine with CD polyethylene glycol hydrogel and Cytosine phosphate Guanine (CpG) nano adjuvant. Nano adjuvants promote antigen presentation and enhance immunity by co-delivering antigens and adjuvants. The hydrogel as a scaffold provides a better growth environment for DCs and recruits endogenous DCs to produce synergy. The vaccine effectively activated DC maturation under the action of CpG nanoparticles. In addition, the combination vaccine significantly increased the infiltration of effector T cells, attenuated the tumor immune suppression microenvironment, and greatly improved the immune effect of the vaccine, providing new ideas for cancer immunotherapy [38] (Figure 5).

Antibody recruitment molecule (ARM) immunotherapy is a promising tumor treatment strategy that has the dual characteristics of recruiting endogenous antibodies and targeting tumors [39,40]. The antibody terminal (ABT) of ARM is usually a hapten such as L-neneneba rhamnose (Rha) or 2,4-dinitrophenol (DNP), which mainly binds to endogenous antibodies in human serum. The tumor binding terminal (TBT) acts as a targeted tumor [41]. Based on this, Zheng et al. reported multivalent antibody recruitment molecules (ARMs) with dual targeting tumor terminal (TBT), including hyaluronic acid (natural ligand targeting CD44) targeting clustered 44 (CD44, a transmembrane glycoprotein, overexpressed in breast cancer and triple negative breast cancer) and nano antibody 7D12 or peptide GE11.7D12 targeting epidermal growth factor receptor (EGFR, overexpressed in tumor cells). GE11 is attached to polyvalent rhamnose via host–guest recognition β- CD-grafted hyaluronic acid (HACD) to form macromolecular complexes. By dual targeting and recruiting Rha antibodies, cytotoxicity against target tumors is generated, providing direction for future ARM strategies with dual specificity [42] (Figure 6). 

### 3.3. Cyclodextrin-Bonded Small Molecule Immunomodulators

Li et al. designed a method to synthesize a “metal-organic skeleton” (γ-CD-MOF) using γ-CD and potassium ions as a backbone, and then modified span-85 with γ-CD-MOF to obtain a novel vaccine adjuvant (SP) with high porosity, good biocompatibility, and a simple green preparation process. SP-γ-CD-MOF was firstly obtained by span-85 modification, followed by the formation of SP-β-CD-MOF/OVA by antigenic model ovalbumin (OVA) loading. The resulting SP-β-CD-MOF/OVA can protect OVA against antigens for sustained release with an enhanced immune response [43].

At present, there are few reports in the literature about CD involved in immunotherapy. However, the remarkable characteristics of CD show its great potential in the field of immunotherapy. CD analogues can be used both as immunotherapeutic agents and as carriers of immune drugs. The unique host–object recognition of CD allows it to better form complexes with immune drugs and better exert the therapeutic effect of immune drugs.

## 4. Combination of Immunotherapy and Other Therapies

### 4.1. Immunotherapy and Chemotherapy

More and more evidence shows that a certain dose of chemotherapy drugs, such as paclitaxel, doxorubicin and cisplatin, can activate the immune system and regulate the tumor microenvironment [44].

Chemotherapy drugs are extensively employed in cancer treatment [45]. Some chemotherapy drugs can induce immunogenic death of tumor cells, which involves the release of antigens from tumor cells, uptake and presentation by DC cells, and activation of cytotoxic T lymphocytes [46,47]. Paclitaxel (PTX) is a commonly used broad-spectrum chemotherapy drug with poor water solubility. Some studies have shown that low-dose PTX can regulate the immunosuppressive tumor microenvironment to improve the immunotherapeutic effect of cytokines. In addition, low-dose PTX can also expose calcium reticulin (CRT) on tumor cells, stimulate DC cells, and rebuild immune monitoring. IL-2 has been widely used as a cytokine approved for melanoma and renal cell carcinoma, but high doses of IL-2 have side effects. Therefore, reducing the dosage of IL-2 may avoid side effects, but low doses of IL-2 alone may not induce immune effector cells, so the binding of PTX and IL-2 is imperative. However, due to differences in physical and chemical properties and mechanisms of action, the two cannot be simply combined directly. Therefore, it is very urgent to develop a drug delivery platform that can simultaneously deliver PTX and IL-2. Song et al. constructed a responsive nano gel for the tumor microenvironment. Specifically, they first obtained methacrylamide N-(2-hydroxy) propyl-3-trimethylammonium chloride chitosan (HTCCm) with opposite charge as a nano gel. By regulating HTCCm, the responsiveness to weak acidic pH was generated. Then, they applied 2-hydroxypropyl-β-cyclodextrin-acrylate (HP-β-CD-A) to the gel system through photo-crosslinking, which can improve the encapsulation efficiency of PTX. In order to encapsulate IL-2 and to achieve the effect of protection and delivery, they further coated the red blood cell membrane (RBCm) on the nano gel (NG). The glycoprotein on the surface of RBCm can better combine with IL-2. This nano gel can better release drugs through response and facilitate a synergistic anti-tumor effect of chemotherapy and immunity [48] (Figure 7).

Chemotherapy drugs can trigger immune responses in the body, but also will be limited by elements of the immune system itself, including indoleamine 2, 3-bis oxygenase 1 (IDO-1) and programmed cell death protein 1 (PD-1) and other factors generated by the immune response [49,50,51]. 

IDO-1 is related to immunosuppression, which can help tumor cells escape detection by the immune system and promote tumor growth and metastasis. At the tumor site, based on CD8+ and interferon γ, IDO-1 is usually overexpressed in the tumor environment [52,53,54]. Tryptophan (Trp) is catalyzed to kynurine (Kyn) under IDO-1 conditions [55]. Trp depletion will adversely affect the survival of CD8+ T cells. Kyn also plays a role in the tumor environment. It activates regulatory T cells and then inhibits the anti-cancer function of cytotoxic T cells (CTL), enabling tumor cells to grow freely without immunosuppression. Therefore, inhibition of IDO-1 has important significance in the immunosuppression of cancer cells. Studies have shown that NLG919 is an imidazoliazionisoindole that can inhibit IDO-1, and it has been proved to be effective in blocking IDO-1mediated immunosuppression [56,57]. However, clinical trials have shown that NLG919 is insoluble in water and cannot be administered by injection [58,59]. 

Hence, it is imperative to have a drug carrier that exhibits facile loading of hydrophobic compounds, while simultaneously possessing commendable biocompatibility and minimal toxicity. CDs and their derivatives manifest exceptional characteristics that fulfill these prerequisites. Their hydrophobic internal cavities effortlessly encapsulate hydrophobic drugs, while the hydrophilic exterior surface enhances water solubility. In light of this, Xu J et al. conducted a comprehensive evaluation of the interaction between various CDs and NLG919, identifying the optimal carrier for NLG919 administration. Among these, HP-β-CD emerged as the most advantageous for facilitating NLG919 dissolution and subsequent intravenous injection. Furthermore, when combined with paclitaxel (PTX), the cytotoxicity toward tumor cells exhibited a substantial augmentation. Notably, the NLG919/HP-β-CD complex demonstrated heightened chemotherapeutic potency in synergy with PTX, thereby attaining immuno-chemotherapeutic synergy and amplifying the overall antitumor efficacy [60] (Figure 8).

An unprecedented “mutual cooperation” between CD and paclitaxel was reported for advanced melanoma therapy. Specifically, a multivalent host–guest interaction composite (pPTX/pCD-pSNO) was constructed by host/guest inclusion complexation between a nitric oxide (pSNO)-modified β-CD polymer and a polymerized paclitaxel (pPTX). In vitro cell experiments confirmed the significant cytotoxicity of this module via inducing immunogenic cell death, activating dendritic cells, and expanding T cells. Further combination with antibodies capable of targeting cytotoxic T lymphocyte antigen-4 led to compromised signal transmission to T cells for enhanced anticancer activity with prolonged animal survival. Compared with an oncolytic virus (OV) approved for the treatment of advanced melanoma, this nano-assembly has the advantages of better safety and controllability. Although OV has been approved for clinical use, it may cause persistent viral infection of normal cells and pose an accidental infection risk to healthcare workers and patient families. Therefore, the developed virus-free nanocomposites represent a promising alternative as improved simulated OV treatment strategies [61] (Figure 9). 

Safiye Akkın et al. devised a CD nanocomposite through charge interaction, aiming to specifically target the widely used chemotherapy drugs fluorouracil (5-FU) and interleukin 2 (IL-2) in the context of cancer treatment. The cationic CD polymers demonstrate a notable affinity for the negatively charged IL-2, thereby facilitating the formation of nanocomposites. Extensive experimental studies have consistently demonstrated that the CD nanocomposites possess a drug loading capacity of approximately 40% for 5-FU and an impressive drug loading capacity of 99.8% for IL-2. Animal studies have revealed the compound’s commendable safety profile in healthy mice, as well as its potent anticancer activity against colon cancer cells in CT26 mice, outperforming the efficacy of 5-FU as a standalone drug. These findings highlight the favorable tumor immunochemotherapy effects achieved through this approach [62] (Figure 10). 

In terms of DOX, Hu et al. reported a GSH responsive supermolecule gold nanocage based on poly cyclodextrin. Because the high GSH state of the tumor microenvironment will lead to the coupling of chemotherapy drugs in the body with GSH, the drugs will then be discharged. The material was first synthesized as a crosslinker (DBHD) corresponding to GSH with p-nitrophenyl carbonate as the terminal group. The CD was covalently crosslinked with the crosslinker DBHD through a disulfide bond, and the GSH-sensitive poly cyclodextrin (PCD) was prepared. DOX can be coupled to the supermolecule gold nanocage of PCD through host–guest recognition. Finally, PEG-NH was introduced to stabilize the system (PDOP). Due to the introduction of PEG-NH, PDOP gold nanocages (NCs) have the characteristics of prolonging blood circulation and improving tumor accumulation. When the gold nanocage enters the tumor site, due to high GSH, the disulfide bond of the gold nanocage will be destroyed, and PDOPNCS will be dissociated. This gold nanocage prevents DOX from expelling tumor through GSH, enhances chemotherapy activity, and induces cell immune death (ICD), thus generating a synergistic effect of immunochemotherapy [63] (Figure 11).

CD also demonstrates its efficacy in the field of natural medicine. Ginsenoside Rg3, derived from ginseng extract, has been identified as a potential inducer of immunogenic cell death (ICD) in tumor cells [64]. Quercetin (QTN), another traditional Chinese medicine ingredient that shares similarities with Rg3, exhibits the ability to generate reactive oxygen species (ROS) and enhance ICD in cancer treatment. To achieve a synergistic immunochemotherapeutic effect using both compounds, Sun et al. developed an amphiphilic CD nano delivery system with folate targeting. This system successfully delivers Rg3 and QTN to the tumor site, thereby exerting their therapeutic effects [65].

In summary, the integration of CD in immunotherapy exhibits commendable performance when synergistically employed with chemotherapy. In essence, the CD-based immunochemotherapy approach holds immense promise for the future of cancer treatment.

### 4.2. Immunotherapy and Photothermal Therapy 

Photothermal therapy (PTT) currently stands as one of the most prominent therapeutic modalities. Noteworthy attributes of this therapy include the absence of systemic toxicity, robust specificity, and remarkable therapeutic efficacy [66]. Photothermal therapy boasts a longstanding history and was predominantly employed in the treatment of skin ailments, including psoriasis and related conditions. The efficacy of photothermal therapy is contingent upon the tissue-penetrating capacity of near-infrared light (NIR). However, the efficacy of photothermal therapy is also constrained by the limitations associated with light penetration. The amalgamation of photothermal therapy with complementary modalities presents a viable avenue for enhancing the therapeutic outcomes of both approaches.

Liu J et al. designed a nanoplatform based on gold nanorods for NIR II-mediated photothermal therapy (PTT). N6 methyladenosine (m6A) demethylase inhibits prostate cancer targeting. For enhanced photothermal immunotherapy, they used polystyrene sulfonate as the interconnection layer, assembled GNR layer by layer, and then used a γ-CD crosslinked low molecular weight polyethylene imide cationic polymer coating, which is related to targeting prostate cancer. An eight-polymer peptide was allowed to bind to a specific gastrin-releasing peptide receptor. Then, the m6ARNA demethylase inhibitor mellofenac (MA) was loaded into the CD cavity through hydrophobic interaction. GNRCDP8MA specifically targets prostate cancer cells and selectively aggregates at tumor sites in vivo. In addition, under 1208 nm laser irradiation, GNR-CDP8MA almost completely eliminated prostate cancer cell-derived tumors. In the mechanism, NIR-II triggers GNRCDP8MA to release MA, which increases the methylation of mRNAm6A and reduces the stability of PD-L1 transcript. In addition, PTT mediated by GNRCDP8MA induces the death of immunogenic cells in primary tumors, thereby enhancing anti-tumor immunity by activating antigen-presenting dendritic cells and tumor-specific effector T cells in metastatic tumors [67] (Figure 12). 

The integration of CD-mediated immunotherapy and photothermal therapy has also found application in cancer vaccine development. Qin L et al. proposed a novel approach by utilizing the photothermal agent indocyanine green (ICG) to encapsulate doxorubicin (DOX), resulting in the α-CD gel system DOX/ICG/CpG-P-ss-M/CD for combined treatment. Upon irradiation, the gel system exhibited controlled release of DOX and vaccine-like nanoparticles CpG-P-ss-M, facilitated by thermal response. The liberated CpG-P-ss-M, acting as a carrier, efficiently adsorbed antigens and facilitated their delivery to lymph nodes (LNs), thereby enhancing antigen uptake and dendritic cell (DC) maturation. Furthermore, when combined with PD-L1 blockade, this therapeutic approach demonstrated remarkable efficacy in suppressing primary tumor growth, inducing tumor-specific immune responses, and combating tumor recurrence and metastasis [68] (Figure 13). 

### 4.3. Immunotherapy and Photodynamic Therapy 

Photodynamic therapy (PDT) has gained significant popularity in recent years as an effective therapeutic approach. Similar to photothermal therapy, PDT also involves the utilization of photosensitizers, but its primary focus lies in the generation of intracellular reactive oxygen species (ROS) [69,70,71]. However, the efficacy of photodynamic therapy is influenced by the penetration of near-infrared light within tissues and cells. To address this limitation, Qi et al. innovatively combined photodynamic therapy with CD immunotherapy to develop an enhanced approach for breast cancer treatment [72]. They engineered a highly efficient subminiature nano reagent (3.1 ± 0.4 nm) composed of polyethylene glycol-modified Cu2x. Through a series of modifications, including the incorporation of selenium nanoparticles and β-CD as well as chloride ion 6, they successfully derived a nano platform known as CS-CDCe6NP. This nano platform demonstrated passive tumor accumulation and exhibited excellent photodynamic therapy effects under near-infrared light irradiation. The abundant production of ROS during PDT not only led to the destruction of primary tumor cells but also triggered immunogenic cell death (ICD) and the release of injury-related factors, thereby facilitating the induction of refined inflammatory M1 macrophages. 

Immunotherapy with PDT and NK cells has shown great potential in addition. However, the activation of NK cells within the tumor microenvironment is inhibited by the shedding of NK group 2, member D ligands (NKG2DLs). Therefore, Liu et al. constructed a microenvironmental, light-responsive bio-nanosystem (MLRN) consisting of β-CD containing SB-3CT and liposomes loaded with photosensitizers. Among them, SB-3CT was suggested to remodel the tumor microenvironment. β-CD and liposomes were linked by a metalloproteinase (MMP-2)-responsive peptide, and sequential release of SB-3CT and dihydroporphine e6 was induced by the MMP-2-enriched tumor microenvironment and 660 nm laser irradiation, respectively. SB-3CT promoted both NKG2D/NKG2DL by antagonizing the MMP-2 pathway to block tumor cell immune escape. This study better integrates PDT and immunotherapy and provides unique insights into the future of PDT and immunotherapy [73].

The combination of multiple therapeutic approaches has been a hot topic in recent years, and the combination of immunotherapy involving CD with other therapies can improve the therapeutic efficacy through synergistic effects while improving the original shortcomings of each therapy. In the future, the combination of CD-involved immunotherapy with other therapies will lead to new discoveries in oncology treatment (Table 2). 

## 5. Conclusions

In recent years, significant progress has been made in the field of immunotherapy, and the use of polymeric materials to mitigate the deficiencies of immunotherapy makes them very popular. The special structure of CD has led to its extensive research in different fields.

In tumor immunotherapy, CD analogues can be used as immune drugs, or they can be functionalized to form complexes with immune drugs that are originally poorly water-soluble, improving the water solubility of immunotherapeutic drugs to enhance their efficacy. In addition, CDs can also be modified to introduce targeting groups, so that immune drugs can bind specifically to target cells and improve the precision of immunotherapy. In addition, CDs can also carry immune escape or immunosuppressive drugs while encapsulating immune drugs, so that the shortcomings of immunotherapy can be improved and the effect of immunotherapy drugs can be enhanced!

The application of cyclodextrins in tumor vaccines also provides a new therapeutic direction for immunotherapy.

The number of articles addressing the application of CDs in immunotherapy is not very large. Therefore, CD-mediated immunotherapy is still in the early stages of development. The pursuit of appropriate antitumor immunotherapeutic approaches is the key to addressing cancer challenges, and the integration of tumor immunotherapy with other therapeutic modalities is imperative. Undoubtedly, CD-mediated immunotherapy has emerged as an effective and promising strategy for antitumor immunotherapy. Looking ahead, the utilization of CDs in immunotherapy has great potential for clinical translation in cancer therapy.

## Figures and Tables

**Figure 1 molecules-28-05610-f001:**
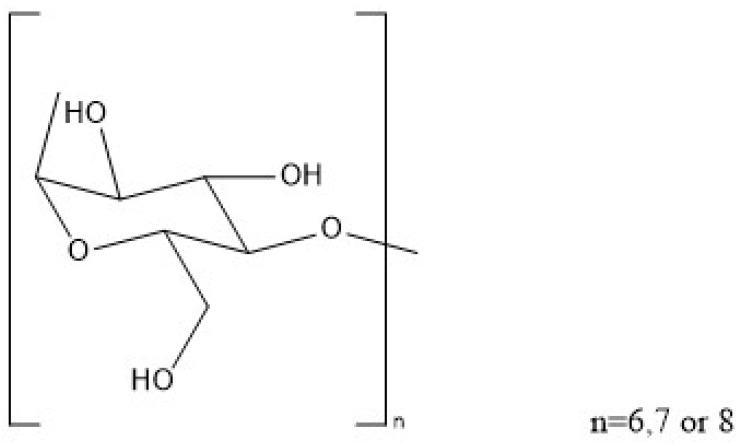
Structure and species of cyclodextrins.

**Figure 2 molecules-28-05610-f002:**
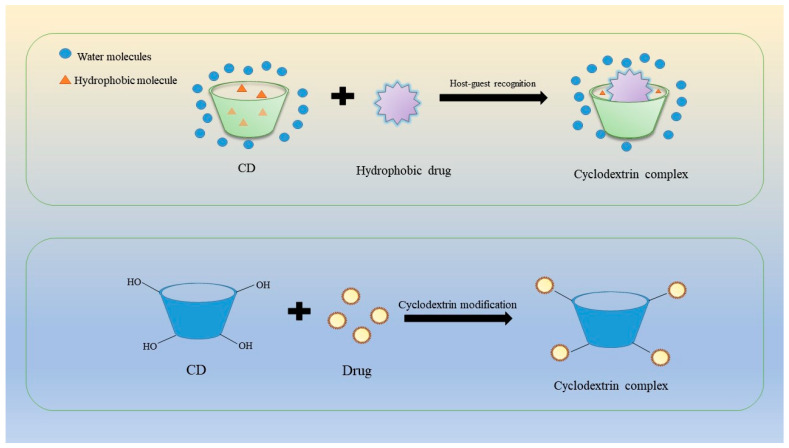
Combination mode of cyclodextrin and drugs.

**Figure 3 molecules-28-05610-f003:**
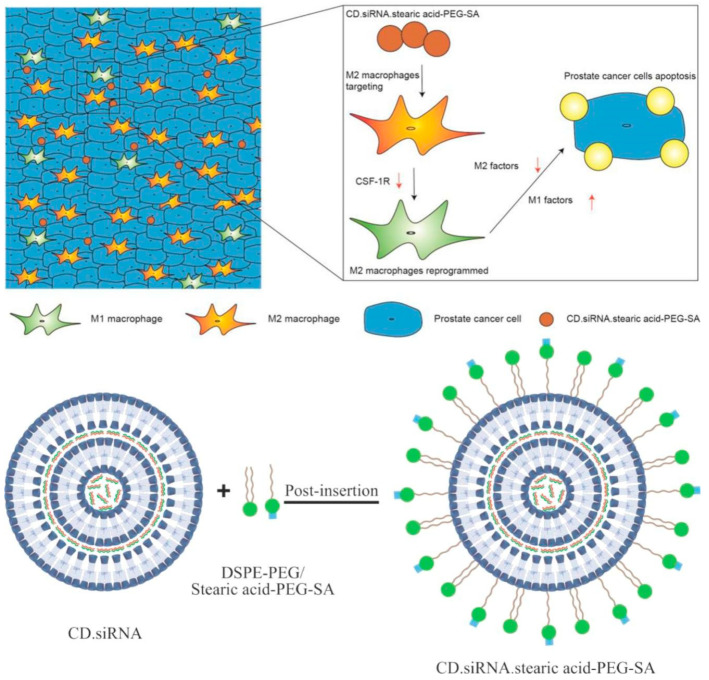
A graphical summary of the pathway (**above**) and a schematic of the formulation of the M2 macrophage-targeting NP CD.siRNA.stearic acid-PEG-SA prepared by incorporating the ligand stearic acid-PEG-SA into CD.siRNA formulation using a post-insertion approach (**below**) (Sun et al. [34]).

**Figure 4 molecules-28-05610-f004:**
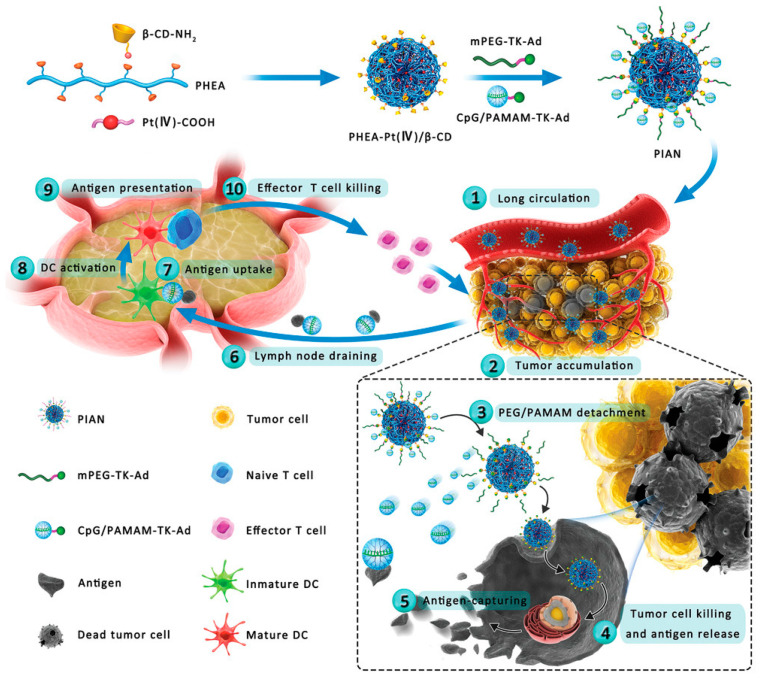
Schematic representation of the programmable immune activation nanomedicine (PIAN) for immune activation and tumor inhibition. PIAN is fabricated through a one-step supramolecular assembly process via beta cyclodextrin (β-CD)/adamantine (Ad) host–guest interactions among various components. Systemically administered PIAN accumulates in the tumor tissue and undergoes dissociation to release poly-[(N-2-hydroxyethyl)-aspartamide]-Pt(IV)/β-CD (PPCD) and CpG/polyamidoamine-thioketal-Ad (CpG/PAMAM). PPCD induces tumor cell death and antigen release, while CpG/PAMAM captures antigens and promotes antigen uptake and dendritic cell (DC) activation. Finally, the activated DCs prime effector T cells for further tumor cell killing (Zhang et al. [37]).

**Figure 5 molecules-28-05610-f005:**
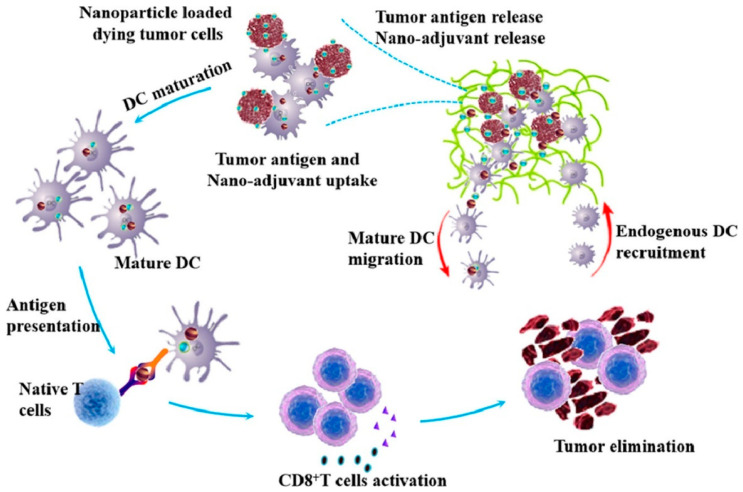
Schematic diagram of hydrogel/nano adjuvant-mediated combined-cell vaccine for cancer immunotherapy (Yang et al. [38]).

**Figure 6 molecules-28-05610-f006:**
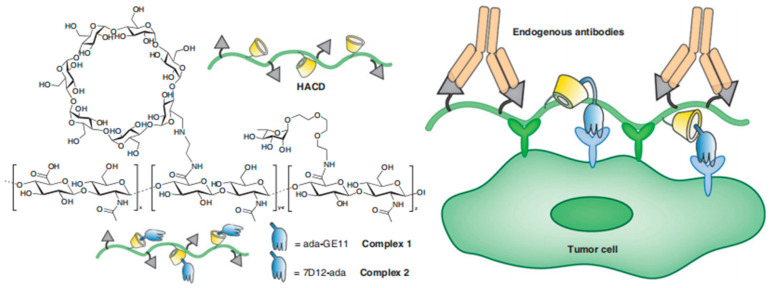
CD44 and EGFR dual-targeted antibody-recruiting complex based on hyaluronic acid grafted with β-Cyclodextrin and multivalent rhamnose for cancer immunotherapy (Zheng et al. [42]).

**Figure 7 molecules-28-05610-f007:**
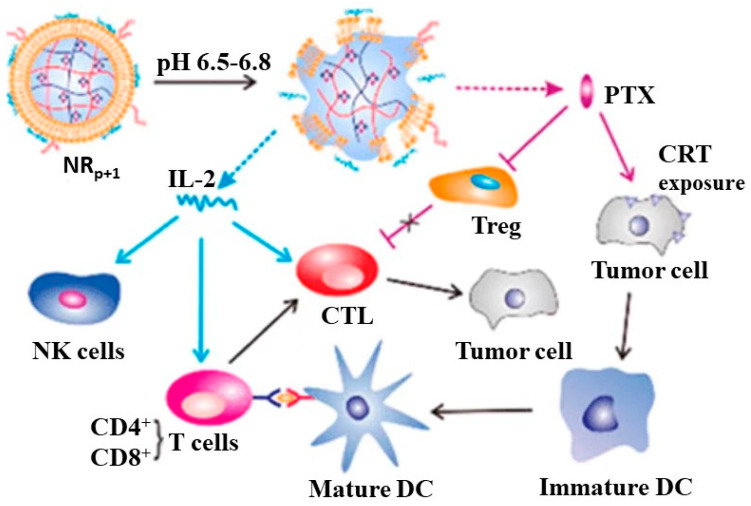
Tumor microenvironment responsive nanogel for the combinatorial antitumor effect of chemotherapy and immunotherapy (Song et al. [48]).

**Figure 8 molecules-28-05610-f008:**
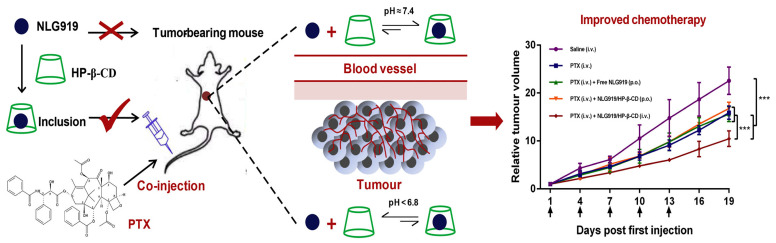
Schematic diagram of NLG919/cyclodextrin complexation combined with paclitaxel, and the relative tumor volume data, *** *p* < 0.001, n = 6. (Xu, J., et al. [60]).

**Figure 9 molecules-28-05610-f009:**
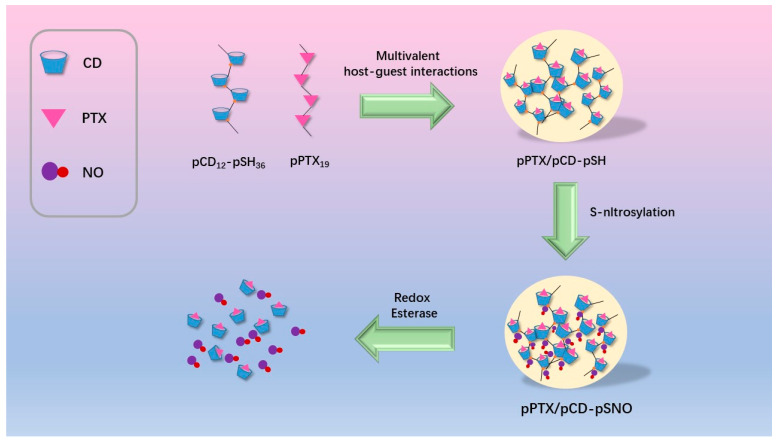
Schematic diagram of Poly(cyclodextrin)-drug nanocomplexes(pPTX/pCD-pSNO). (Kim J., et al. [61]).

**Figure 10 molecules-28-05610-f010:**
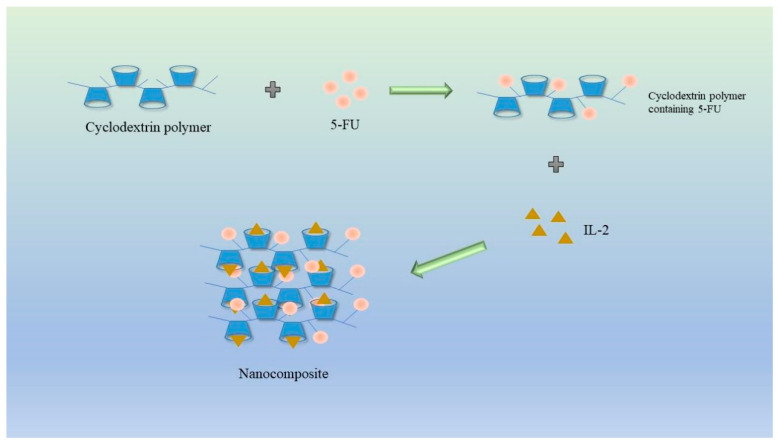
Schematic diagram of cyclodextrin and interleukin-2 nanocomplex loaded with 5-FU for the treatment of colon cancer [62].

**Figure 11 molecules-28-05610-f011:**
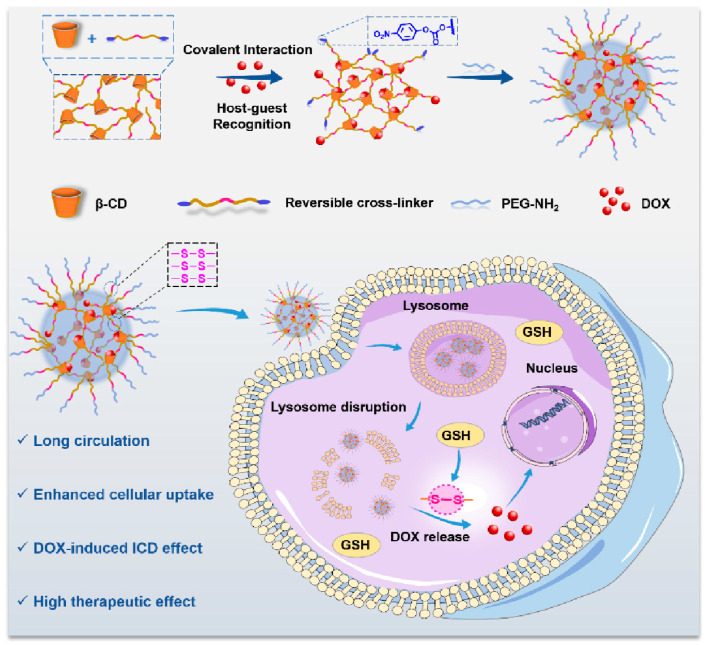
Schematic diagram of reduction triggered polycyclodextrin supermolecule gold nanocage (Hu et al. [63]).

**Figure 12 molecules-28-05610-f012:**
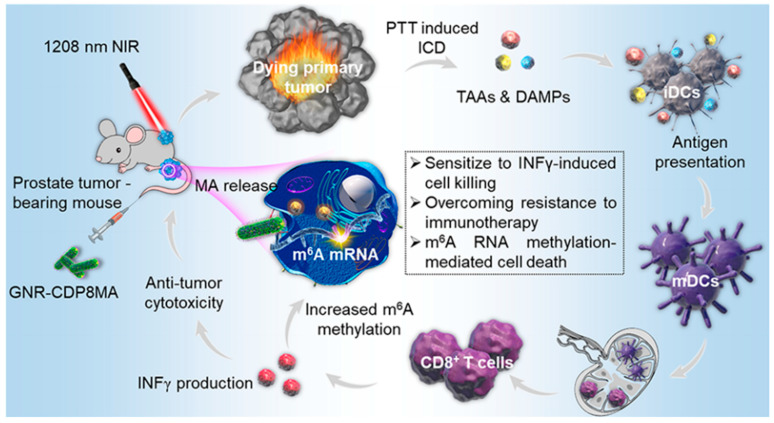
Cyclodextrin-functionalized gold nanorods loaded with meclofenamic acid for improving N6-methyladenosine-mediated→second→near-infrared photothermal immunotherapy (Liu J., et al. [67]).

**Figure 13 molecules-28-05610-f013:**
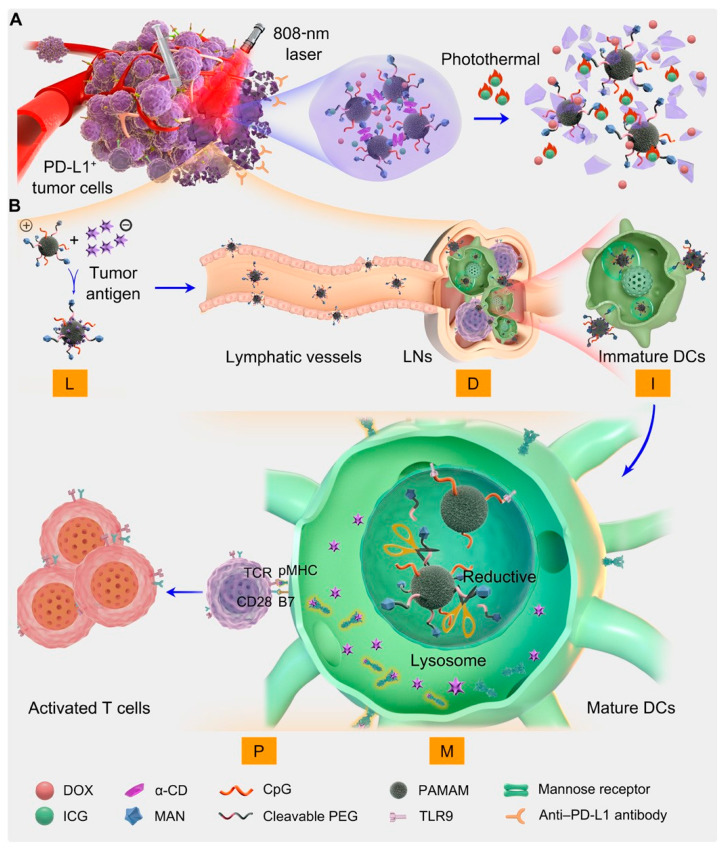
(**A**) Fabrication of the integrated regimen and the release process of CpG-P-ss-M. (**B**) Simplified mechanism of CpG-P-ss-M–mediated DC maturation for cancer immunotherapy. Letters LDIMP in orange frame represent loading tumor-specific antigens by DDS, draining to LNs, internalization by DCs, DC maturation for costimulatory molecule expression, and presenting peptide–MHC-I complexes to T cells, respectively. (Qin, L. et al. [68]).

**Table 1 molecules-28-05610-t001:** Characteristics of several cyclodextrins.

Cyclodextrin	Molecular Formula	M_W_ [g/mol]	Solubility in Water at Room Temperature [mg/mL]	Hydrogen Bond Donor Count	Hydrogen Bond Acceptor Count
α-CD	C_36_H_60_O_30_	972	130	18	30
β-CD	C_42_H_70_O_35_	1132	18.4	21	35
γ-CD	C_48_H_80_O_40_	1297	249	24	40

**Table 2 molecules-28-05610-t002:** Advantages of cyclodextrin-based materials for cancer immunotherapy.

**Advantages of Cyclodextrin**	**Reference**
Cyclodextrin analogues as immunotherapeutic drugs	[32]
Host–guest recognition with drugs for enhanced bioavailability	[34,37,38,42,48,60,61,62,63,65,66,67,68,73]
Easy to modify on the primary and secondary faces for active targeting	[34,42,65]
Induction of a strong immune stimulus	[38]
Coordination with alkali metals	[43]
Cationic CD polymer-mediated electrostatic interactions	[62]

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
