# Peer review of "Application of Cyclodextrin for Cancer Immunotherapy"

_molecules, 2023, doi:10.3390/molecules28145610_

Round 1

Reviewer 1 Report

This review summarized recent advances in cyclodextrin based materials for cancer immunotherapy. cyclodextrin-mediated immunotherapy as demonstrated by the remarkable examples above, due to the complexity of tumor therapy, cyclodextrin alone cannot produce immunotherapy effects; it must be combined with immunodrugs to achieve immunotherapy However, some categories of polymer hybrids are too rough, and the content is not finely summarized. I will reconsider my suggestion after you submit a new revision.

1.     For first-time reviews proposed by authors, authors are advised to maintain objectivity; there are already many relevant reviews on CD, such as:Progress in Polymer Science, 2019, 93: 1-35. Progress in Polymer Science, 2021, 118: 101408. and it is recommended to avoid using terms such as first-time.

2.     The statement of the problem should not appear as a section but should be unified into INTRODUCTION more appropriately.

3.   The authors have listed three CDs; however, they have not provided an explanation on how these materials differ concerning their biomedical application. To address this gap, it would be beneficial to include several sentences that details the distinctive features and potential applications of each CD material.

4.     The meaning of Figure 2 does not clearly convey the actual drug content and the picture needs to be optimized.

5.     For the review paper, more informative pictures are a must, images also need to be copyright compliant for citation

6.     For comparing the drug delivery effects, some related research about the drug delivery polymers should be cited to highlight the potential applications of CD. Pharmaceutics, 2023, 15(2): 368.; Biomolecules 2022, 12, 636.

Reviewer 2 Report

The paper overall is poorly written and I struggled to understand what the paper is trying to achieve. The introduction identifies the core concept of focusing on the use of cyclodextrins in cancer immunotherapy. It states this is due to challenges with immunotherapy including poor water solubility and immune escape of drugs. However no references or supporting evidence is provided for this.

Section 2 Current status of immunotherapy is poorly written, very basic and in places inaccurate. It again also does not in any way focus on highlighting the specific issues that were called out in the introduction i.e. solubility/immune escape. This should be the focus of this section so as you can then show how cyclodextrins are addressing these limitations. 

Some examples of poor writing/inaccuracy

Tumor immunotherapy is to achieve the anti-tumor effect through the body's own immune system, which is a persistent immune immune response

Innate immunity is mainly composed of T lymphocytes, B lymphocytes and antigen-presenting cells. - ?

section 2.2.2. It is a peptide vaccine consisting of immunogenic epitopes, usually derived from tumor-specific or tumor-associated antigens 

section 2.2.3Cytokines are some chemicals produced by the immune system[26]. Cytokines play an important role in the production of immune cells and blood cells.

Section 3 - again content is very vague and non-specific, and lacks focus to the actual topic of the paper.You talk about antiviral, anti inflammatory and anticancer - but in essence what all 3 of these sections are focusing on is the same thing from a physicochemical perspective i.e. increasing solubility/bioavailability? What is the relevance of each of these sections to the overall purpose of the paper? It feels like all these different sections were written from a general perspective and then just put together without ensuring that the overall paper has a clear message and structure 

Example

Viruses are an important factor that endangers human life[50-52]. Every year, more than 50 million people around the world die from viral infections. In response to viral infec-tions, scientists have devoted their efforts to the development of new chemical entities, medicinal biomolecules and vaccines. - how does this relate to the paper being written?

Section 4 - 4.1. is titled 'cyclodextrin as immunomodulators' but then the conclusion to that paragraph is that it is not an immunomodulator? This section was also very hard to understand - it needs to be rewritten in a much clearer manner. 

section 4.2 - another example of a very broad generalisation 

Interleukin-2 (IL-2) is a cytokine in metastatic melanoma, which can inhibit melanoma, but its response is limited. - is IL-1 specific to metastatic melanoma? It is not but the first commercial cytokine approved for clinical use was an IL-2 in metastatic melanoma. That is very different to what you have said.

The section on the paper by Park et al., makes no sense. 

The section on the paper by McBride et al, is interesting but again it actually only focuses on the role of cycodextrin in a very minor way. 

Section 4.3 At the end of this section the paper states 'The reason may be that the therapeutic effect of the combination of cyclodextrin and single immunotherapy is not ideal.' which somewhat undermines the aim of the paper. 

Section 5 - again the focus seems to be primarily on describing immunotherapeutic mechanisms with limited emphasis on cyclodextrins other than to say they increase solubility - but what does this achieve? This is already widely known and alot of the focus seems to be on the effect of cyclodextrin on standard chemotherapeutics and how by enhancing their bioavailability their activity (in some cases on the immune system) is enhanced but again I do not feel this is adding anything new to the literature.

Reviewer 3 Report

We congratulate the authors for the study and for choosing this research topic of major interest for the medical field.

We agree with the publication of the manuscript in the Molecules Journal, only after performing the following revisions as follows:

- completing the Abstract part by mentioning the immunotherapy strategies described in the article and the perspective of optimizing the therapeutic efficacy when associated with cyclodextrins;

- page 1 – first paragraph – correct form – metastasis

- page 1 – first paragraph – correct form – such

- page 1 – at the end of the first paragraph – to cite some examples of drug molecules that have low solubility in water and immune escape of tumor cells

- explanations for all abbreviations at their very first entry within the text (see IDO – page 2, Fv region and Fc region – page 2, DOX – page. 6, β-Rapa – page 8)

- mark with italics in vitro and in vivo for the entire paper (Latin expressions)

- page 1 – last paragraph – correct form – which is a persistent immune response

- page 2 – second paragraph – correct form – Cellular immunity is

- page 5 – correct form – 3.2.2. Anti-inflammatory

- page 5 – correct form – diabetes, etc.

- page 8 – correct form – surface area. The

- page 9 – correct form – (Trp) is catalyzed

- page 11 – correct form – to penetrate tissue. However,

- page 13 – correct form – cells, respectively. (Qin L et al.).

References should be described as follows, depending on the type of work:

· Journal Articles:

1. Author 1, A.B.; Author 2, C.D. Title of the article. Abbreviated Journal Name Year, Volume, page range.

· Books and Book Chapters:

2. Author 1, A.; Author 2, B. Book Title, 3rd ed.; Publisher: Publisher Location, Country, Year; pp. 154–196.

3. Author 1, A.; Author 2, B. Title of the chapter. In Book Title, 2nd ed.; Editor 1, A., Editor 2, B., Eds.; Publisher: Publisher Location, Country, Year; Volume 3, pp. 154–196.

· Unpublished materials intended for publication:

4. Author 1, A.B.; Author 2, C. Title of Unpublished Work (optional). Correspondence Affiliation, City, State, Country. year, status (manuscript in preparation; to be submitted).

5. Author 1, A.B.; Author 2, C. Title of Unpublished Work. Abbreviated Journal Name year, phrase indicating stage of publication (submitted; accepted; in press).

· Unpublished materials not intended for publication:

6. Author 1, A.B. (Affiliation, City, State, Country); Author 2, C. (Affiliation, City, State, Country). Phase describing the material, year. (phase: Personal communication; Private communication; Unpublished work; etc.)

· Conference Proceedings:

7. Author 1, A.B.; Author 2, C.D.; Author 3, E.F. Title of Presentation. In Title of the Collected Work (if available), Proceedings of the Name of the Conference, Location of Conference, Country, Date of Conference; Editor 1, Editor 2, Eds. (if available); Publisher: City, Country, Year (if available); Abstract Number (optional), Pagination (optional).

· Thesis:

8. Author 1, A.B. Title of Thesis. Level of Thesis, Degree-Granting University, Location of University, Date of Completion.

· Websites:

9. Title of Site. Available online: URL (accessed on Day Month Year).

Unlike published works, websites may change over time or disappear, so we encourage you create an archive of the cited website using a service such as WebCite. Archived websites should be cited using the link provided as follows:

10. Title of Site. URL (archived on Day Month Year).

Round 2

Reviewer 1 Report

1)It is recommended that authors provide a table to compare and demonstrate the advantages of CD materials.

2) For comparing the drug delivery effects, some related research about the drug delivery polymers should be cited to highlight the potential applications of CD. Pharmaceutics, 2023, 15(2): 368.; Biomolecules 2022, 12, 636.Biomacromolecules 2021, 22, 2, 732–742
